# Characterisation of HILDCAA events using Recurrence Quantification Analysis

Odim Mendes[1], Margarete Oliveira Domingues[2], Ezequiel Echer[1], Rajkumar Hajra[3], and Varlei Everton Menconi[1]

[1]Space Geophysics Division (DGE/CEA), Brazilian Institute for Space Research (INPE), São José dos Campos, São Paulo, Brazil.
[2]Associated Laboratory of Computation and Applied Mathematics (LAC/CTE), Brazilian Institute for Space Research (INPE), São José dos Campos, São Paulo, Brazil.
[3]Laboratoire de Physique et Chimie de l'Environnement et de l'Espace (LPC2E), CNRS, Orléans 45100, France.

*Correspondence to:* Odim Mendes (odim.mendes@inpe.br)

**Abstract.** Considering the magnetic reconnection and the viscous interaction as the fundamental mechanisms for transfer particles and energy into the magnetosphere, we study the dynamical characteristics of Auroral Electrojet ($AE$) index during High--Intensity Long--Duration Continuous Auroral Activity (HILDCAA) events, using a long-term geomagnetic database (1975-2012), and other distinct interplanetary conditions (geomagnetically quiet intervals, Co-rotating Interaction Regions (CIRs)/High Speed Streams (HSS) not followed by HILDCAAs, and events of $AE$ comprised in global intense geomagnetic disturbances). It worths noting that we also study active but non-HILDCAA intervals. Examining the geomagnetic $AE$ index, we apply a dynamics analysis composed of the phase space, the recurrence plot (RP), and the Recurrence Quantification Analysis (RQA) methods. As a result, the quantification finds two distinct clustering of the dynamical behaviours occurring in the interplanetary medium: one regarding a geomagnetically quiet condition regime and the other regarding an interplanetary activity regime. Furthermore, the HILDCAAs seem unique events regarding a visible, intense manifestations of interplanetary Alfvenic waves; however, they are similar to the other kinds of conditions regarding a dynamical signature (based on RQA), because it is involved in the same complex mechanism of generating geomagnetic disturbances. Also, by characterising the proper conditions of transitions from quiescent conditions to weaker geomagnetic disturbances inside the magnetosphere and ionosphere system, RQA method indicates clearly the two fundamental dynamics (geomagnetically quiet intervals and HILDCAA events) to be evaluated with magneto-hydrodynamics simulations to understand better the critical processes related to energy and particle transfer into the magnetosphere-ionosphere system. At last, with this work, we have also reinforced the potential applicability of the RQA method for characterising of nonlinear geomagnetic processes related to the magnetic reconnection and the viscous interaction affecting the magnetosphere

# 1 Introduction

A complicated electrodynamic region populated by plasmas and ruled by the Earth's magnetic field — designated in a classical definition as magnetosphere — exists surrounding our planet (Mendes et al., 2005; Kivelson and Russell, 1995). This region is exposed to influences of the space environment and submitted to several interplanetary forcings. Initially, a summary view of the physics scenario involved is briefly described in the two following paragraphs.

In electrodynamic terms, three main solar agents (*i* - electromagnetic radiation, *ii* - energetic particles and *iii* - solar magnetized structures) act upon the Earth's atmosphere, which is permeated by a magnetic field created in the interior of our planet (Campbell, 2003; Hargreaves, 1992). *(i)* Electromagnetic radiation both heats the planet globally and ionises the atmosphere. This ionisation gives basis to a terrestrial plasma environment. *(ii)* Also, the incidence episodes of solar energetic particles increase the ionisation in a manner much more localised. *(iii)* Furthermore, escaping in a continuous way from the Sun, the solar wind, superposed sometimes by coronal mass ejections structures and other peculiar solar structures (e.g., solar fast speed streams and heliospheric current sheet), transports intrinsically the solar magnetic field until the orbit of the Earth and beyond (Kivelson and Russell, 1995). Two primary electrodynamic interactions are possible from this incidence of the magnetised solar wind plasma upon the Earth's magnetosphere. These interactions result in a transfer of energy and particles into the magnetosphere boundary. The most intense is through the magnetic reconnection process (Burch and Drake, 2009; Kivelson and Russell, 1995; Dungey, 1961), when the Interplanetary Magnetic Field (IMF) presenting a predominantly southward orientation, in the Geocentric Solar Magnetosphere reference system, merges into the geomagnetic field at the outer boundary and produces strong modification in a large region formed by the magnetosphere and the ionosphere, the latter is a region about 100 to 2000 km of altitude presenting the highest quantity of ionized particles. Another competitive process is the Kelvin-Helmholtz viscous interaction (Hasegawa et al., 1997; Chen et al., 2004; Axford and Hines, 1961). Most of the time this second process is in action when the magnetosphere acts as a closed physical system, concerning the incident frontal solar wind, due to an IMF with northward orientation. A macroscopic fluid dynamics developed by the plasma sliding at the flanks of the magnetosphere creates a kind of viscous interaction, which produces the mixing of the solar plasma inside the magnetosphere and the occurrence of ULF waves (Menk and Waters, 2013) affecting the interior regions. The former process is more efficient in energy and particle transfer than the latter one.

In a global sense, during events of solar wind transporting IMF parallel (northward) to the frontal geomagnetic field, a regime of low magnetic disturbance on the ground is noticed. However, when the IMF is strongly southward directed, antiparallel to the geomagnetic field, intense regimes of disturbances are recorded on the ground. Nevertheless, there is a peculiar interplanetary process related to manifestations of Alfvèn waves (Guarnieri et al., 2006), presenting alternation of the magnetic component orientation (in the southward-northward direction), which produces an intermediate level of geomagnetic disturbance with the typical duration of days. These nonlinear Alfvén waves are known to be the main origin of high-intensity long-duration continuous Auroral Electrojet ($AE$) activity (HILDCAA) events on the Earth (Hajra et al., 2013; Tsurutani et al., 2011b, a; Echer et al., 2011; Tsurutani et al., 1990; Tsurutani and Gonzalez, 1987). As presented in Davis and Sugiura (1966), the $AE$ is a geomagnetic index related to the quantification of the geomagnetic disturbance produced by enhanced

ionospheric electric currents flowing below and within the auroral region (https://www.ngdc.noaa.gov/stp/geomag/ae.html). The primary mechanism for these HILDCAA events is the high-speed solar wind streams (HSSs) emanating from solar coronal holes accompanied by embedded Alfvén waves (Belcher and Davis, 1971; Tsurutani et al., 1994), which are characterised by significant IMF variability (see Echer et al. (2012, 2011); Tsurutani et al. (2011b, a)). The sporadic magnetic reconnection

(Dungey, 1961; Gonzalez and Mozer, 1974) formed between the southward component of the Alfvén waves and the Earth's magnetopause fields leads to intense substorm/convection events comprising HILDCAAs (Tsurutani et al., 1995), which are shown to last from days to weeks (Tsurutani et al., 1995, 2006; Gonzalez et al., 2006; Guarnieri, 2006; Kozyra et al., 2006; Hajra et al., 2013, 2014a). The HILDCAA events carry a large amount of solar wind kinetic energy input into the magnetosphere affecting the polar ionosphere (Gonzalez et al., 2006; Hajra et al., 2014b). More than 60% of this energy is dissipated

in the magnetosphere-ionosphere system. Additionally, other importance of these events is the accelerated relativistic electrons, known as killer electrons, in the Earth's radiation belts (Hajra et al., 2014c, 2015b, a) for their hazardous effects on orbiting spacecraft (Wrenn, 1995; Horne, 2003). The variations of $AE$ during HILDCAAs show the nonlinear dynamics of the physical processes involved. Therefore, a dynamical characterisation is of fundamental interest for a deeper insight into the electrodynamic coupling between the solar wind and the magnetosphere related.

The aim of this work is to highlight dynamical characteristics related to the HILDCAA events revealed by the $AE$ index in the context of the electrodynamic coupling processes. With this purpose, we apply here then the phase space analysis, the recurrence plot (RP) technique, and the recurrence quantification analysis (RQA) method (Eckmann et al., 1987; Maizel and Lenk, 1981; Trulla et al., 1996). They constitute proper tools to treat such nonlinear, non-stationary signals as in Geophysics processes. Such analysis method is applied to the HILDCAA events, for the first time to our knowledge, allowing a comparison of

dynamical characteristics. By applying the nonlinear tools, this work investigates $AE$ under some distinct physical conditions of the interplanetary medium: Alfvenic fluctuations followed with HILDCAA, Alfvenic fluctuations not followed by HILDCAA (also related to Co-rotating Interaction Regions (CIRs) and high-speed streams (HSS)), other disturbed interplanetary conditions, and geomagnetically quiet time.

This work presents the content as follows. Section 2 describes the methods for analysis. Section 3 presents the geomagnetic

database and how we apply the methodology. Section 4 shows the results and interpretations. At last, Section 5 summarise the conclusions.

## 2 Method of Analysis

The information theory structures a branch of powerful mathematical tools to analyse nonlinear systems of signal proposed in the seminal paper of the mathematician Claude E. Shannon (Shannon, 1964). An analogy with the concept of entropy

from Physics gives basis to these tools. As reviewed and discussed in details by Cover and Thomas (2006), the entropy $H$ used as basis for the methods can be expressed by $H(\mathbf{X}) = -\sum P(x)\log(P(x)), \quad x \in \mathbf{X}$, where $\mathbf{X}$ is the set of all messages $\{x_1, ..., x_n\}$ that $\mathbf{X}$ could be, and $P(x)$ is the probability of some $x \in \mathbf{X}$. In this work, we use quantification methods associated with this theory, precisely the method developed by Zbilut and Webber Jr. (1992) of Recurrence Quantification Analysis (RQA)

that is built from the Recurrence Plot (RP), as introduced in Eckmann et al. (1987), and the proprieties of the phase space, provided in *Cross Recurrence Plot Toolbox* [1]. Initially, these methods are used to analyse dynamical systems from a theoretical

point of view. Nevertheless, since the late 90's, they have been extended to experimental data to characterise nonlinear complex behaviour (Trulla et al., 1996; Marwan and Webber, 2015). Below we present the phase space, the RP and the RQA approaches briefly.

### Phase space

A phase plot is a geometric representation of the trajectories of a dynamical system in the phase plane. It is a fundamental

starting point of many approaches in nonlinear data analysis, which is based on the construction of a phase space portrait of the considered system. A review on that can be found, for instance, in N. Marwan's tutorial[2]. The state of a system can be expressed by its state variables $x_1(t), x_2(t), ..., x_d(t)$, for instance, the state variables density, pressure, momentum, and magnetic field for a magneto-hydrodynamics system. The $d$ state variables at time $t$ establish a vector in a $d$-dimensional space which is called phase space. The state of a system changes in time, and, consequently, the vector in the phase space describes a

trajectory representing the time evolution, i.e. the dynamics of the system. Accordingly, the appearance of the trajectory retains information about the system. Therefore, the phase space is formed by coordinates that represent each significant variable of the system to specify an instantaneous state (Marwan, 2003).

In practice, observations of a real process do not unveil all state variables, or they are not known, or they can not be measured. Nevertheless, due to the couplings between the system components, we can reconstruct a phase space trajectory from a single

observation by a time delay embedding (Takens, 1981). It yields to the so called Takens' embedding theorem, which states that a reconstruction of the phase space trajectory $\boldsymbol{x}(t)$ from a time series $u_k$, with a cadence $\Delta t$, allows to present a proper dynamics of a system. In order to do that, an embedding dimension $m$ and a time delay $\tau$ must be identified, related to the following reconstruction: $\boldsymbol{x}(i) = \boldsymbol{x}_i = (u_i, u_i i + \tau, ..., u_{i+(m-1)\tau})$, where $t = i\Delta t$. Here, $m$ is found by using the false nearest neighbour method and $\tau$ by the mutual information method (Kennel et al., 1992; Marwan and Webber, 2015). The idea

behind this approach is to identify the influence of increasing the embedded dimension $m$ in the number of neighbours along a trajectory of the system.

### Recurrence Plot (RP)

The Recurrence Plot is based on the Poincaré's recurrence theorem from 1890, as discussed in Schulman (1978). It states that a dynamic system returns to a state arbitrarily close to the initial state after a particular time. Mathematically the RP is

obtained by the square matrix $\mathbf{R}_{i,j} = \Theta(\epsilon_i - \| \boldsymbol{x}_i - \boldsymbol{x}_j \|)$, where $\epsilon_i$ is a predefined cut-off distance, $\| . \|$ is the norm (in our case, the Euclidean norm) and $\Theta(x)$ is the Heaviside function (Eckmann et al., 1987). The binary values zero and one in this matrix are represented by white and black creating visual patterns.

The characteristic typology (related to macro patterns) and texture (related to micro details) presented in the RP are the key points of the interpretation. However, the visual interpretation of RPs requires some training experience, usually done

---

[1]Cross Recurrence Plot Toolbox 5.21 (R31b) by the Interdisciplinary Center for Dynamics of Complex Systems, University of Potsdam (http://tocsy. pik-potsdam.de/CRPtoolbox/)

[2]http://www.agnld.uni-potsdam.de/~marwan/matlab-tutorials/html/phasespace.html#13.

from standard systems or data libraries . For instance, as described in Marwan et al. (2007) and in the RP and RQA website http://www.recurrence-plot.tk: (*i*) stationary processes are associated to homogeneous distribution of points in RP; (*ii*) periodic processes present cycle patterns where the distance between periodic patterns corresponds to the period; (*iii*) long diagonal lines with different distances to each other reveal a quasi-periodic process; (*iv*) non-stationary processes can present interruption on the lines, this can indicate as well some rare state, or RP fading to the upper left and lower right corners indicating also trend or drifts; (*v*) single isolate points demonstrate heavy fluctuation in the process, in particular if only isolate points occur an uncorrelated or anti-correlated random process are represented; (*vi*) evolutionary processes are illustrated by diagonal lines, then the evolution of states is similar at different times. However, if it has parallel lines related to the main diagonal, the system is deterministic (or even chaotic, if they occur beside single lines), and if the diagonal lines are orthogonal to the main diagonal, or the time is reversed or the choice of embedding is insufficient; (*vii*) long bowed line structures express evolution states that are similar at different epochs although they have different velocity (the dynamics of the system could be changing); (*viii*) vertical and horizontal lines/clusters are evidences that the states has no or slow change for some time, which point to a laminar state.

The establishment of quantifiers to express the characterisation of the processes described in RP was a significant advance in the popularisation of this tool, because it can help to express in a concise and objective way a description on the dynamics of the processes, as discussed in Marwan and Webber (2015) and reference therein. Therefore, quantification from RP come primarily from the recurrence patterns, and it presents such as point density, diagonal structures, and vertical structures in the recurrence plot. On the following text, we present four of these quantifiers to study the behaviour of physical conditions such as geomagnetically quiet intervals and HILDCAA cases.

**Recurrence Quantification Analysis (RQA)**

Trulla et al. (1996) addressed the problem of quantifying the structure that appears in the RPs and used them to analyse experimental data. This approach is useful to reveal qualitative transitions in a system. The corresponding measurements capture the dynamical characters of the system as represented by the signal. Therefore, the RQA measures provide a qualitative description of a system regarding complexity measures (Marwan et al., 2007). We refer to Marwan and Kurths (2002), and Marwan (2003) to a detailed discussion on this subject. Notably, the diagonal structures in the RP and the recurrence point density are used to measure the complexity of a physical system (Zbilut and Webber Jr., 1992; Webber Jr. and Zbilut, 1994). In the present work we restrict our analysis to four characteristic parameters described below:

1. **Recurrence rate (RR)**: It denotes the overall probability that a certain state recurs and it is obtained from the RP by $RR = \sum_{i,j=1}^{N} \frac{R_{i,j}(\rho)}{N^2}$. Larger values, more recurrence.

2. **Determinism (DET)**: It represents how predictable a system is, and it is expressed by the ratio of recurrence points that form diagonal lines of the RP of at least length $\ell_{min}$ to all recurrence points, *i.e.,* $DET = \frac{\sum_{\ell=\ell_{min}}^{N} \ell P(\ell)}{\sum_{\ell=1}^{N} \ell P(\ell)}$, where $P(\ell)$ denotes the probability to find a diagonal line of length $\ell$ in the RP.

3. **Laminarity (LAM)**: It measures the occurrence of laminar states and it is related to intermittent regimes, namely, it is the ratio between the recurrence points forming the vertical lines and the entire set of recurrence points computed by

$LAM = \frac{\sum_{\nu=\nu_{min}}^{N} \nu P(\nu)}{\sum_{\nu=1}^{N} \nu P(\nu)}$, where $P(\nu)$ denotes the probability to find a vertical line of length $\nu$ in the RP. LAM does not describe the length of laminar phases, however if this measure decreases the RP consists of more single recurrence points than vertical structures. This measurement is relatively more robust against noise in signals.

     4. **Entropy (ENT)**: It reflects the complexity of the deterministic structure in the system referred to Shannon entropy (Shannon, 1964), namely, $ENT = -\sum_{\ell=\ell_{min}}^{N} p(\ell) \ln(p(\ell))$, where $p(\ell) = P(\ell)/N_\ell$. This measure reflects the complexity of

the RP concerning the diagonal lines. In this form computed from RP, the interpretation of these values are different of traditional Shannon entropy, *i.e.*, larger values are related to low entropy compared to physics analogy (Letellier, 2006).

## 3   Database and Methodology Procedure

For the present work, we have considered an updated list of 136 HILDCAA events occurring between 1975 and 2012, compiled by Hajra et al. (2013). The events were detected from the geomagnetic $AE$ and, middle to low latitude disturbance, $Dst$ indices

by using the four strict HILDCAA criteria (Tsurutani and Gonzalez, 1987): (i) the events have peak $AE$ intensities greater than 1000 nT, (ii) the events last for more than 2 days, (iii) high auroral activity lasts throughout the interval, i.e., $AE$ never drops below 200 nT for more than 2 h at a time, and (iv) the events take place outside of the main phase of a geomagnetic storm. To a better understanding, the main phase is determined by the depression in the horizontal component, from middle to low latitudes, in the geomagnetic field. This behaviour is identified and quantified using the hourly value equatorial $Dst$ index,

which represents ideally the axially symmetric disturbance magnetic field at the dipole equator on the Earth's surface. This index is derived monitoring the equatorial ring current variations (http://wdc.kugi.kyoto-u.ac.jp/dstdir/dst2/onDstindex.html). The $AE$ dataset is provided by the OMNIweb Service (http://omniweb.gsfc.nasa.gov/) by NASA and $Dst$ from World Data Center for Geomagnetism, kyoto $Dst$ index service (http://wdc.kugi.kyoto-u.ac.jp/dstdir/).

    From the list, the first 16 events were eliminated due to incomplete information. Among the remaining events, 33% were

preceded by geomagnetic storm main phase ($Dst < -50$nT). Thus 80 events were analysed in this work, because we selected the events classified as pure HILDCAAs, i.e., events not preceded by any geomagnetic storm main phase.

    As data sets, the high time-resolution (one minute) $AE$ indices were analysed to study the dynamical characterisation of the HILDCAA events by the RQA method. To eliminate any marginal influences, we considered a 2280 minute interval centred at the middle point of a HILDCAA event. This number of records was determined by the least interval among the events.

For a quantitative comparison of disturbance geomagnetic regimes, we also performed the same RQA during the geomagnetically quiet period listed in Table 1. The quiet days follow the criteria: $Kp \leq 3^0$, $Dst \geq -50$ nT, and $AE \leq 300$ nT. The planetary three-hour-range $Kp$ index was introduced by J. Bartels in 1949 and designed to be sensitive to any geomagnetic disturbance affecting the Earth (http://www.gfz-potsdam.de/en/section/earths-magnetic-field/data-products-services/kp-index/ explanation/). It completes a set of indices to diagnose the level of geomagnetic disturbance in a global sense. The geomagnetic indices (Rostoker, 1972) can be obtained from Word Data Center, Kyoto, in http://wdc.kugi.kyoto-u.ac.jp/wdc/Sec3.html. That way, if any, different physical regimes undoubtedly allow us to find a distinct characterisation of the signals. In our case, we investigate periods of HILDCAA events that alter a physical regime that exists during the geomagnetically quiet times.

For a more complete dynamical diagnosis, this work investigates $AE$ index under some other different physical conditions of the interplanetary medium. Completing the earlier mentioned cases of the interplanetary Alfvenic fluctuations followed by HILDCAA (related to CIRs and HSS) and the geomagnetically quiet time, cases of interplanetary Alfvenic fluctuations not followed by HILDCAA (also related to CIRs and HSS) and cases of intense interplanetary conditions (characterised by simultaneous activities in the AE, Dst and Kp indices) produced by different interplanetary causes are also analysed. Table 2
presents the CIRs/HSS not followed by HILDCAA event. The first column shows the data set interval and the second column the 2280-minute interval considered in the analysis calculations. Table 3 presents the events with $AE$ index related to global intense geomagnetic disturbances. The first column shows the data set interval and the second column the 2280-minute interval considered in the analysis calculations.

The analyses of the results allow a comparison of the dynamical characteristics of signals.

**4   Results**

Initially, two typical cases are shown and analysed, one from the HILDCAA events and another from the quiet time intervals. As examples for the methodology application, they help to understand the analysis and its interpretation. Fig. 1 shows $AE$ variations including a HILDCAA interval. The HILDCAA started at 1734 UT on 30 May (day 150) and continued till 0934 UT on 02 June (day 153) of 1986, with a total duration of about 64 hours. In that figure, the double arrow horizontal line indicates
the exact interval of the event. For the RQA calculation we consider the 2280-minute interval centred at the middle of the HILDCAA. Two vertical dotted lines mark this interval. Fig. 2 shows $AE$ variations during a geomagnetically quiet period. The plot shows the geomagnetically quiet period from 17 to 22 July (day 198 to day 203) of 2006 (from Table 1). The region between the two vertical dotted lines shows the same 2280-minute interval selected to the RQA study as in the HILDCAA case.
From the $AE$ plots, the differences in the amplitudes between the HILDCAA interval (peak about $1200$ nT) and the quiet time interval (peak about $300$ nT) are remarkable, as expected. Both of them presents fluctuations in the signal intensities. The application of the RQA methodology aims at characterising the dynamical behaviour of the signals.

Fig. 3 represents the phase space plots for the HILDCAA. As a value estimated by the earlier mentioned mutual information methodology, the time delay ($\tau$) used is $34$ minutes. The phase space charts present snapshots of the interconnections of the
records for each case. As described by the theory in Section 2, the geometric representation in the plot gives the trajectory of the dynamical system involved in the $AE$ index records. Although slightly insinuated by the distribution of points, a proper representation is not achieved because the noise in the signal disturbs the identification of the trajectory. Following the same procedure, Fig. 4 gives the representation for the quiet interval shown earlier. The time delays ($\tau$) found is also $34$ minutes. Although the signal amplitude is quite different compared to the one of the HILDCAA event, the trajectory behaviour is almost similar. A question arises from the comparison, is it possible to distinguish from the dynamical behaviour analyses the two kinds of occurrences as the $AE$ indices point out?

To verify whether the question deserves the study efforts, we use the RP technique to allow a visual inspection of the signal features. Dealing with the RP theory for all the cases studied, we estimated the typical values related to these dynamical systems. The embedded dimension ($m$) determined by the false nearest neighbour method was found to be around 6, and following the time delay ($\tau$) was around 34 minutes. The cut-off distance ($\epsilon$) was $\approx 30\,\mathrm{nT}$ for the HILDCAAs, and $\approx 10\,\mathrm{nT}$ for the quiet intervals. For the other interplanetary conditions, the values were similar to the value of HILDCAAs. The estimation of $\epsilon$ uses a value defined by the additive effects of the data resolution and the Gaussian noise threshold.

Related to the cases at the beginning of this section, Fig. 5 shows the RPs for the HILDCAA and Fig. 6 for the quiet interval. Here we take the embedded dimension ($m$) and the time delay ($\tau$) equal to 1 for RQA calculations. These parameter choices take into account the categorisation purpose of the present work, and those values do not alter our characterisation process (Iwanski and Bradley, 1998; March et al., 2005; Marwan, 2011). The RPs highlight the recurrences in the signal records showing differences in the dynamical patterns between the HILDCAA interval and the quiet period. To the both systems, the analyses on the large scale patterns in the plots, designated as typology, denote that they are of the disrupted kind, i.e., with abrupt changes in the representation of the dynamics. However, the analysis of the small scale patterns, designated as texture, denotes a more complex dynamics in the HILDCAA event than the one in the quiet interval. To obtain an objective interpretation, we need to translate this visual appreciation to quantitative descriptors of the dynamics of the system interpreted by the $AE$ index. As examples of this quantification, the results of the RQA dynamical parameters for the quiet and HILDCAA case examples are presented in Table 4. We verify they are about one order of magnitude smaller for the HILDCAA than the values for the quiet interval. That way, we have a shred of evidence that encourages this kind of study.

To pursue a comprehensive answer, we apply the RQA methodology to all 80 HILDCAA events completed by the examination of other cases selected (6 geomagnetically quiet intervals, 5 CIRs/HSS not followed by HILDCAA, and 4 events of $AE$ in global intense geomagnetic disturbances) to allow comparisons. The values of the RQA dynamical variables (RR, DET, LAM, and ENT) were obtained for each case.

Here, Table 5 shows the minimum, maximum, mean, standard deviation, median, and mode values estimated to the HILDCAAs and the quiet periods. As it can be seen, a difference of one order of magnitude for each variable exists between these cases. For minima and maxima, the differences are between half and one order of magnitude. The standard deviation, median, and mode are in agreement with normal distributions for the phenomena.

At last, Fig. 7 shows the RQA dynamical parameters for all events under study. For each parameter, we normalised the values for all events concerning extreme values obtained for the parameter. The empty circles represent the HILDCAA events, and the plus signs show the quiet periods. A clear distinction between the HILDCAA events and quiet time intervals may be noted from the figure. The separation of the results for the HILDCAA event and the quiet time interval establishes a clustering of the results, which characterise two well defined physical regimes. Further, the symbol **x** indicates the results for $AE$ index in CIR/HSS events not followed by HILDCAA, and **\*** in a whole global disturbance scenario. As also seen in the figure, parameter behaviour are similar for CIR/HSS causing HILDCAAs and CIR/HSS not causing them, and distinct of the behaviour of quiet intervals. Therefore, based on this plot, one could say that the bottom part shows the behaviour of Alfvenic solar wind intervals, CIR and HSS; while the top part shows the behaviour related to the slow solar wind interval. The analysis

taking into account the $AE$ in a whole global disturbance scenario regarding geomagnetic behaviour shows larger spreading values for the parameters except by the RR parameter; nevertheless, values are also different of the one in quiet time regime. Based on the current geophysical knowledge, the RQA patterns in the signals for these events help to characterise/identify the standard physical features. Examining the physics of every case in the active interplanetary regimes, one might point out that $AE$ signature relates to HILDCAA that is connected to long duration, large amplitude Alfvenic fluctuations; to CIR/HSS not followed by HILDCAA connected to short term Alfvenic fluctuations and with or without a small interplanetary southward magnetic amplitude; and to events in a global geomagnetic disturbance scenario connected to small amplitude southward interplanetary magnetic field without Alfvenic fluctuations or to a large southward interplanetary magnetic amplitude.

Thus, the RQA-result comparisons allows to achieve some interpretations.

The HILDCAAs seem unique events regarding a visible, intense manifestations of interplanetary Alfvenic waves; however, they are similar to the other kinds of conditions regarding a dynamical signature (based on RQA), because it is involved in the same complex mechanism of generating geomagnetic disturbances.

Supplying basis to the geomagnetic disturbances, mainly the $AE$ studied here, the physics scenario could be properly interpreted according to a basic view. As known, the fundamental mechanisms are the magnetic reconnection and viscous interaction with a transfer of energy and particles by electrodynamics interaction and generation of geomagnetic disturbance on ground. Supported by the parameter clustering behaviours shown in the Fig. 7, the interpretation obtained from the RQA examination of $AE$ index is in agreement with those fundamental mechanisms. Although describing an expected result, the quantitative study using this method indicates in a clear way categories of phenomena (showed in Fig. 7). On the one hand, during geomagnetically quiet conditions, the effective interaction is the ram pressure on the solar front side of the magnetosphere and the development of viscous interaction at flanks. On the other hand, during HILDCAA events, the two fundamental electrodynamics interactions (magnetic reconnection and viscous interaction) with a transfer of energy and particles are indeed happening. In principle, interplanetary phenomena producing both of those coupling mechanisms, as processes examined in (Ma et al., 2014), concern the mechanisms related to interplanetary Alfvén waves. In this kind of occurrence, magnetic disturbances can be detected by magnetometers at the polar regions as the HILDCAA events. Although they can be clearly noticed at high latitudes, those disturbances are noticed as weak worldwide manifestations. CIR/HSS occurrences not followed by HILDCAA related to short term Alfvenic fluctuations and with or without small southward interplanetary magnetic amplitude produce sporadic, low $AE$ index disturbances, designated as geomagnetic substorms. Events in a whole global disturbance scenario related to large southward interplanetary magnetic amplitude produce geomagnetic storms and associated geomagnetic substorms.

Identified as distinct regimes by the RQA diagnose, the geomagnetically quiet intervals and HILDCAA events seem the proper conditions of transitions from quiescent conditions to weaker geomagnetic disturbances inside the magnetosphere and ionosphere system. So, those RQA features can be useful for other study purposes. RQA method gives a clear indication of the dynamics to be evaluated by magneto-hydrodynamics simulations, as developed by Ma et al. (2014) or Chen et al. (2004), to understand the processes involved in a transfer of energy and particles into the magnetosphere-ionosphere system.

## 5  Conclusions

Obtained from a diagnose of features of a nonlinear system analysis, a physics scenario of the Auroral Electrojet ($AE$) index is built with the aid of the Recurrence Quantification Analysis (RQA) information extracted from the Recurrence Plot (RP) calculation. We performed this analysis using 80 HILDCAA events completed by the examination of other cases selected (6 geomagnetically quiet intervals, 5 CIRs/HSS not followed by HILDCAA, and 4 events of $AE$ in global intense geomagnetic disturbances) to allow comparisons.

Some significant RQA variables (RR, DET, LAM and ENT) quantify and characterise the dynamical signatures of the $AE$ index related to HILDCAA occurrences and other interplanetary environment conditions.

A summary of the key findings are:

– The quiet intervals as compared to HILDCAA intervals are characterized by larger values of DET, LAM and ENT, which means higher predictability, lower entropy and larger laminarity of the corresponding nonlinear dynamics.

– There are distinct clustering, identified by RQA, of the dynamical behaviours recorded on the ground produced by the interplanetary medium conditions: one regarding a geomagnetically quiet condition regime and another regarding an effective disturbed interplanetary regime.

– The RQA results identify similar dynamical behaviours for HILDCAA events and the other else disturbed cases.

– On the one hand, the HILDCAAs seem unique events regarding the visible, intense manifestations of Alfvenic waves; on the other hand, however, they are similar to the other else regarding dynamical signatures (based on RQA), because they are involved in the same complex mechanism of generating geomagnetic disturbances.

– This complex mechanism is composed by the magnetic reconnection and the viscous interaction implying ground geomagnetic effects triggered by the southward interplanetary magnetic field.

– One regime of clustering is $AE$ index organised by geomagnetically quiet conditions, related to a predominant interaction from the incidence of ram pressure on the solar front side of the magnetosphere and the development of viscous interaction at flanks, while there is a northward interplanetary magnetic field (IMF). Another regime is $AE$ organised by disturbed interplanetary conditions, with the presence of the southward IMF.

As the geomagnetically quiet intervals and HILDCAA events characterise the proper conditions of transitions from quiescent conditions to weaker geomagnetic disturbances inside the magnetosphere and ionosphere system, RQA method gives a clear indication of the two fundamental dynamics to be evaluated with magneto-hydrodynamics simulations to understand in a better way the fundamental processes related to energy and particle transfer into the magnetosphere-ionosphere system.

With the present work, we have also demonstrated the potential applicability of the RQA method for characterising of nonlinear geomagnetic processes related to magnetic reconnection and viscous interaction affecting the magnetosphere, mainly with the aid of magneto-hydrodynamics simulations.

*Author contributions.* All authors discussed the idea and the approach for the work development and took part in the preparation of the manuscript. O. Mendes and M. O. Domingues worked also in the application of the methodology.

*Competing interests.* The authors declare that they have no conflict of interest.

*Acknowledgements.* M.O.D. and O.M. thank the MCTIC/FINEP (CT-INFRA grant 0112052700) and FAPESP (Grant $2015/25624 - 2$) for the financial support. O.M., M.O.D. and E.E. thank the Brazilian CNPq agency (grants $312246/2013 - 7$, $306038/2015 - 3$ and $301233/2011 - 0$, respectively). R.H. thank the FAPESP $2012/00280 - 0$ for a postdoctoral research fellowship at INPE, and now the work supported by ANR under the financial agreement ANR-15-CE31-0009-01 at LPC2E/CNRS. V.E.M.thank the MCTIC-PCI program (Grant $455097/2013 - 5$) by the research fellowship at INPE. The authors would like to thank the team of Interdisciplinary Center for Dynamics of Complex Systems, University of Potsdam, for the RQA tools (tocsy.pik-potsdam.de), OMNIweb Service (http://omniweb.gsfc.nasa.gov/) by NASA and the World Data Center for Geomagnetism, Kyoto, Japan (http://wdc.kugi.kyoto-u.ac.jp/), where the geomagnetic indices used in this study were collected from. The authors thank Olga Verkhogfolyadova and another anonymous referee for constructive and useful suggestions leading to significant improvement of the manuscript.

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

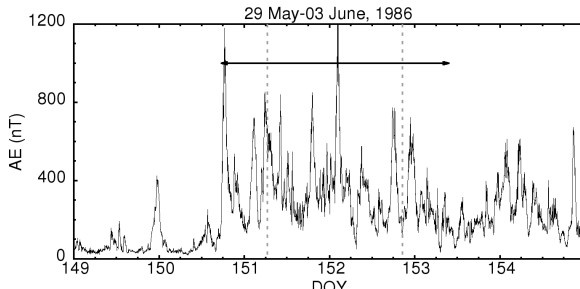

**Figure 1.** Geomagnetic $AE$ index from 29 May (DOY 149) to 03 June (154) 1986 includes a HILDCAA event. The HILDCAA interval is identified by the double arrow horizontal line, and the $AE$ interval used for the RQA is shown between the vertical dotted lines.

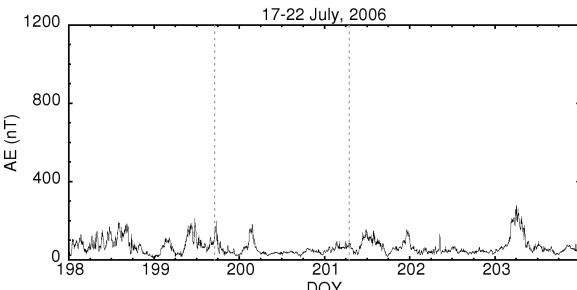

**Figure 2.** Geomagnetic $AE$ index during the geomagnetically quiet period on 17 (DOY 198) - 22 (203) July 2006. The $AE$ interval used for the RQA is marked by vertical dotted lines.

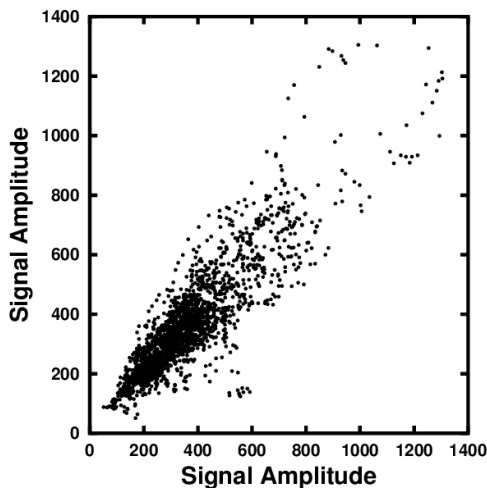

**Figure 3.** The phase space representation for the HILDCAA example shown in Fig. 1. The delay time is 34 minutes.

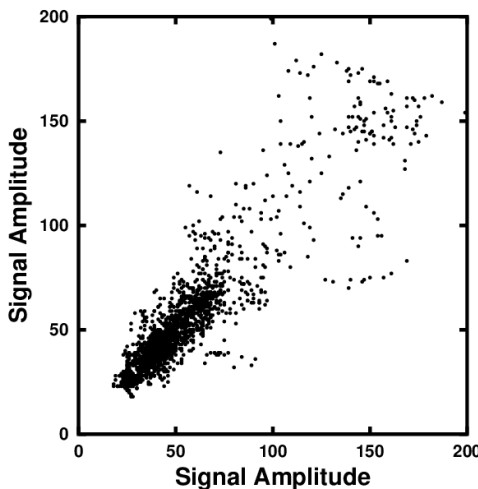

**Figure 4.** The phase space representation for the geomagnetically quiet period example shown, between the vertical dotted lines, in Fig. 2. The delay time is 34 minutes.

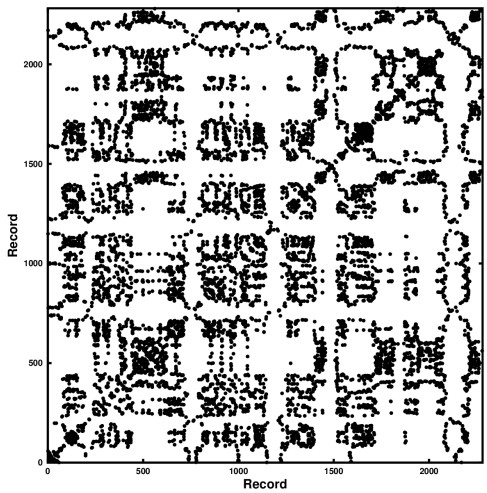

**Figure 5.** The recurrence plot for the HILDCAA example.The interval shown by the vertical dotted lines in Fig. 1 is used to obtain the plot.

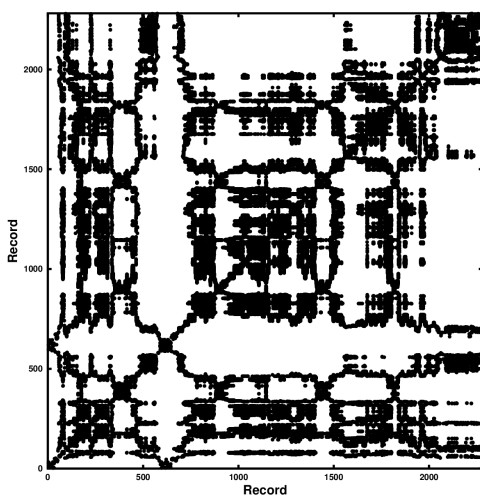

**Figure 6.** The recurrence plot for the geomagnetically quiet period example.The interval shown by the vertical dotted lines in Fig. 2 is used to obtain the plot.

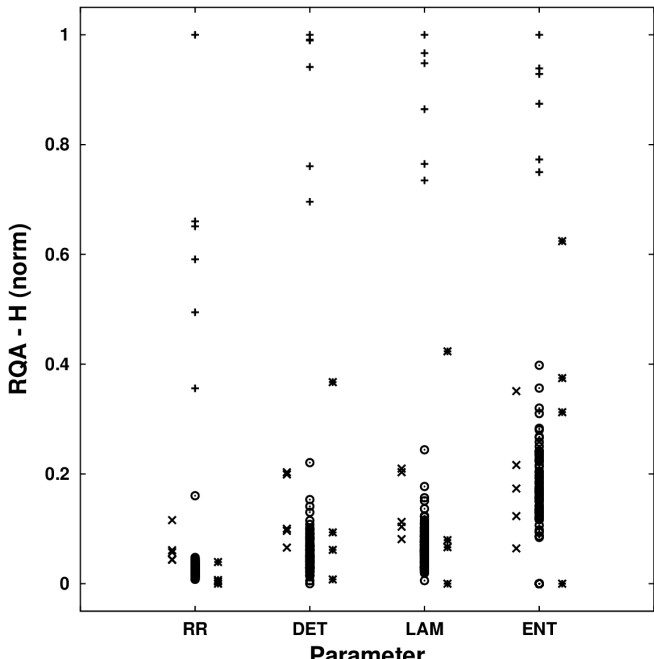

**Figure 7.** Normalized representation of the RQA parameters for Auroral Electrojet $AE$ indices in HILDCAA events (○),in CIR/HSS not followed by HILDCAA (x), in a global geomagnetic disturbance scenario (*), and in the geomagnetically quieter intervals (+).

**Table 1.** The geomagnetically quiet intervals

| Date | $Kp \leq$ | $AE \leq$ | $Dst \geq$ |
|---|---|---|---|
| 14 - 18 November 2000 | $3^0$ | $267\,\text{nT}$ | $-20\,\text{nT}$ |
| 26 - 30 November 2001 | $3^-$ | $133\,\text{nT}$ | $-50\,\text{nT}$ |
| 19 - 25 June 2004 | $2^0$ | $167\,\text{nT}$ | $0\,\text{nT}$ |
| 19 - 27 June 2006 | $2^0$ | $167\,\text{nT}$ | $-9\,\text{nT}$ |
| 15 - 23 July 2006 | $2^0$ | $200\,\text{nT}$ | $32\,\text{nT}$ |
| 01 - 09 December 2007 | $3^0$ | $200\,\text{nT}$ | $-5\,\text{nT}$ |

**Table 2.** CIRs/HSS not followed by HILDCAA

| Data set interval | Interval considered |
|---|---|
| 2008, 012-018 (Jan. 12 to 17) | 2008, Jan. 14 (00:00h)) - 15 (13:59h) |
| 2008, 030-036 (Jan. 30 to Feb. 04) | 2008, Feb. 02 (00:00h)) - 03 (13:59h) |
| 2008, 058-064 (Feb. 27 to Mar. 03) | 2008, Mar. 02 (00:00h)) - 03 (13:59h) |
| 2008, 165-171 (Jun. 13 to 18) | 2008, Jun. 15 (00:00h)) - 16 (13:59h) |
| 2008, 175-181 (Jun. 23 to 28) | 2008, Jun. 26 (00:00h)) - 27 (13:59h) |

**Table 3.** AE in global intense geomagnetic disturbances

| event | Interval considered |
| :---: | :---: |
| 2012, (Mar. 09) | 2012, Mar. 09 (00:00h)) - 10 (13:59h) |
| 2012, (Apr. 23-24) | 2012, Apr. 23 (00:00h)) - 24 (13:59h) |
| 2012, (Jun. 17) | 2012, Jun 17 (00:00h)) - 18 (13:59h) |
| 2012, (Jul. 15) | 2012, Jun 15 (00:00h)) - 16 (13:59h) |

**Table 4.** RQA measures for the Geomagnetically quiet interval and typical HILDCAAS cases

| Case | RQA Measures | | | |
|---|---|---|---|---|
| | RR | DET | LAM | ENT |
| Geomagnetically quiet interval | 0.0203 | 0.357 | 0.518 | 0.719 |
| HILDCAA period | 0.0021 | 0.044 | 0.069 | 0.147 |

**Table 5.** The Recurrence Quantification Analysis results considering two typical cases

| | HILDCAA period | | | | Geomagnetically quiet interval | | | |
|---|---|---|---|---|---|---|---|---|
| Value | RR | DET | LAM | ENT | RR | DET | LAM | ENT |
| Min | 0.0010 | 0.010 | 0.014 | 0.000 | 0.0115 | 0.251 | 0.397 | 0.574 |
| Max | 0.0056 | 0.086 | 0.139 | 0.273 | 0.0307 | 0.357 | 0.536 | 0.766 |
| Mean | 0.0016 | 0.031 | 0.049 | 0.091 | 0.0195 | 0.321 | 0.473 | 0.672 |
| Std | 0.0005 | 0.012 | 0.020 | 0.073 | 0.0065 | 0.046 | 0.058 | 0.075 |
| Med | 0.0015 | 0.028 | 0.046 | 0.104 | 0.0194 | 0.345 | 0.487 | 0.690 |
| Mod | 0.0013 | 0.010 | 0.014 | 0.000 | 0.0115 | 0.251 | 0.397 | 0.574 |