# Peer review of "Characterisation of HILDCAA events using Recurrence Quantification Analysis"

_Nonlinear Processes in Geophysics, 2016_

## Referee Comment (RC1) · Anonymous Referee #1 · 3 Jan 2017

This paper has many weaknesses: the methodology is not explained in sufficient detail, the conclusions are simply not understandable, the English is very poor.

Section 2 should be devoted to explain the mathematical method. Instead, it is merely a list of nomenclature and definitions. How is the Shannon entropy used in the paper? How are the four parameters defined? What do we learn from them?

Section 4 presents the result in a very hurried and superficial way. On line 29, page 5 the Authors argue that the behaviour in Figures 1 and 2 are very similar. They look very different to me. Why should they be similar? One is storm time, the other is quite time!

I do not understand how are Figures 3 and 4 generated and what they represent.

Figures 5 and 6 show the RP matrix, but what do we learn from them?

On line 16-17, page 6 we learn the RQA parameters calculated for the two cases, but again I have no idea what they mean, without reading their definition and physical interpretation.

Finally, I simply cannot make any sense of the three conclusions on page 7. The first one seems wrong: it cannot be that auroral activity is responsible for energy transfer from the solar wind to the magnetosphere-ionosphere. The causality relationship is obviously in the opposite direction!

Regarding the second and third conclusions I do not argue that they are wrong. I just do not understand what they are supposed to mean.

Despite all my criticism, I support the idea of using methods from dynamical systems and chaos theory to analyse geomagnetic events. It should be done, however, in a much more clear and accessible way. As it stands, this paper would not be understood/appreciated by the largest majority of the community.

---

## Referee Comment (RC2) · Anonymous Referee #2 · 5 Jan 2017

This paper opens an interesting possibility to apply the RQA to analyze dynamic properties of auroral activity in HILDCAA or in response to different heliospheric drivers. Unfortunately, the paper does not provide physical interpretation of the results. Specifically, physical meaning of information presented in Figures 5 and 6 is not discussed. I have a problem with the claim that dynamic properties of HILDCAA-driven activity are "unique". There are distinct differences with the quiet-time study, but uniqueness can be shown only in comparison with other non-HILDCAA driven auroral activity. For instance, analysis of CIR/HSS storms without HILDCAA could be helpful. I encourage the authors to continue and extend the study with the focus on understanding physical processes behind different types of auroral activity.
2016.

---

## Author Comment (AC2)

[revised manuscript text omitted]

---

## Author Response (AR1)

**Characterisation of HILDCAA events using Recurrence Quantification Analysis**
by
Mendes et al.

**Reviewer #1**
**comments and answers**

Referee = **R** and Authors' answer = **A**

**General comments of the authors:**
Initially, we thank the Reviewer for the suggestions. A PDF of the paper revised is attached and presents the complete information (In it, the color blue indicates parts of the text revised or even sections completely revised).

**R:** This paper has many weaknesses: the methodology is not explained in sufficient detail, the conclusions are simply not understandable, the English is very poor.
**A:** We rewrote the text taking into account all the remarks. We added details in the RP and RQA explanations in the methodology and some new references to clarify some points. We also revised the content according to the suggestions taking care of language issues.

**R:** Section 2 should be devoted to explain the mathematical method. Instead, it is merely a list of nomenclature and definitions. How is the Shannon entropy used in the paper? How are the four parameters defined? What do we learn from them?
**A:** We rewrote this section to clarify the measure definitions. The patterns presented in the RP can give a qualitative interpretation of the complexity of data being analysed. For that, with some practice, we can observe the patterns described in the vertical/diagonal/horizontal lines, structures and clusters, isolated points, and the corner of the matrix. While a resource to a visual inspection, the RP technique is still an important tool in the complexity analysis. However, the RQA measurements provide an objective quantification of RP based on the structures presented on it. RQA becomes a powerful tool to characterise complex non-linear data, helping us to analyse the dynamical system under investigation on the data.
Entropy (ENT) refers to ideas presented in the Shannon entropy according to Shannon(1964). This measure reflects the complexity of the recurrence plot with respect to the diagonal lines. It represents the probability to find exactly a diagonal line of length $\ell$ in the RP. Nevertheless, the interpretation of the values of this measure is the opposite of traditional Shannon entropy, *i.e.*, larger ENT values are related to low entropy, as presented in Letellier(2006). In the general sense, the concept of entropy is the basis for the quantitative aspects of the Information Theory, translated by the researchers in this area in a mathematical formalism to build non-linear analysis tools for applying, for instance, to physics dataset.

**R:** Section 4 presents the result in a very hurried and superficial way. On line 29, page 5 the Authors argue that the behaviour in Figures 1 and 2 are very similar. They look very different to me. Why should they be similar? One is storm time, the other is quite time!
**A:** The Section 4 was rewritten to improve the presentation of the results and their interpretations. On the mentioned figures the focus was the variability feature, not to the signal intensity. Now, it is presented taking into account both the variability and the amplitude.

**R:** I do not understand how are Figures 3 and 4 generated and what they represent.
**A:** They represent a space phase plot (portrait), that is a geometric representation of the trajectories of the dynamics of the system in the phase plane. The fundamental starting point of many approaches in non-linear data analysis is the construction of a phase space portrait of the considered system. The state of a system can be described by its state variables $x_1(t), x_2(t), ..., x_d(t)$, for example the both state variables temperature and pressure for a thermodynamic system. The "$d$" state variables at time $t$ form a vector in a $d$-dimensional space which is called phase space. The state of a system typically changes in time, and, hence, the vector in the phase space describes a trajectory representing the time evolution, the dynamics, of the system. The shape of the trajectory gives hints about the system; periodic or chaotic systems have characteristic phase space portraits.
The observation of a real process usually does not yield all possible state variables. Either not all state variables are known or not all of them can be measured. However, due to the couplings between the system's components, we can reconstruct a phase space trajectory from a single observation $u$ by a time delay embedding, as described in Takens' embedding Theorem, 1981. Takens proved that instead of $2m + 1$ generic signals, the time-delayed versions

$$u(t), u(t - \tau), u(t - 2\tau), \cdots , u(t - 2m\tau),$$

of one generic signal would suffice to embed the m-dimensional manifold. There are some technical assumptions that must be satisfied, restricting the number of low-period orbits with respect to the time-delay $\tau$ and repeated eigenvalues of the periodic orbits. The phase space reconstruction is not exactly the same to the original phase space; nevertheless, its topological properties are preserved, if the embedding dimension is large enough (the embedding dimension has to be larger than twice the phase space dimension). Details can be found in N. Marwan webpage `http://www.agnld.uni-potsdam.de/~marwan/matlab-tutorials/html/phasespace.html`.

**R:** Figures 5 and 6 show the RP matrix, but what do we learn from them?

**A:** Quantification measurements from RP come basically from the recurrence patterns it presents, such as point density, diagonal structures, and vertical structures in the recurrence plot. By visual inspection, we can see the signatures (from typology and texture) of the processes under consideration. We compare the dynamical behaviour of processes to identify features and, maybe, similarities. We rewrote the text of Section Results to improve the content and the flux of ideas.

**R:** On line 16-17, page 6 we learn the RQA parameters calculated for the two cases, but again I have no idea what they mean, without reading their definition and physical interpretation.

**A:** We rewrote the text to improve the understanding in this part.

**R:** Finally, I simply cannot make any sense of the three conclusions on page 7. The first one seems wrong: it cannot be that auroral activity is responsible for energy transfer from the solar wind to the magnetosphere-ionosphere. The causality relationship is obviously in the opposite direction!

**A:** We improved the way we present the ideas in this section.

**R:** Regarding the second and third conclusions I do not argue that they are wrong. I just do not understand what they are supposed to mean.

**A:** We rewrote the text (with the major points) to improve the understanding in this part.

**R:** Despite all my criticism, I support the idea of using methods from dynamical systems and chaos theory to analyse geomagnetic events. It should be done, however, in a much more clear and accessible way. As it stands, this paper would not be understood/ appreciated by the largest majority of the community.

**A:** We improved the readability of the text to clarify the points the Reviewer presented.

**Reviewer #2**
**comments and answers**

Referee = **R** and Authors' answer = **A**

**General comments of the authors:**
Initially, we thank the Reviewer for the suggestions and encouragement. A PDF of the paper revised is attached and presents the complete information (In it, the color blue indicates parts of the text revised or even sections completely revised).

**R:** This paper opens an interesting possibility to apply the RQA to analyze dynamic properties of auroral activity in HILDCAA or in response to different heliospheric drivers. Unfortunately, the paper does not provide physical interpretation of the results.

**A:** This aspect of the paper has been improved to contribute to the application of the non-linear methodology and the interpretation of the results obtained. Dealing with the integration of three areas (Space Physics, data analysis, and methodology of non-linear science), the authors revisited the paper adding more details (in all sections) to provide an attractive and acceptable work.

**R:** Specifically, physical meaning of information presented in Figures 5 and 6 is not discussed.

**A:** The characteristics of the typology and texture present in the RP are the key points of the interpretation; however, the visual interpretation of RPs requires some training experience, usually done from standard systems or data libraries. For instance, as described in Marwan et al. (2007) and in the RP and RQA website `http://www.recurrence-plot.tk`:

(*i*) stationary processes are associated with a homogeneous distribution of points in RP;

(*ii*) periodic processes present cycle patterns (check board) where the distance between periodic patterns corresponds to the period;

(*iii* long diagonal lines with different distances to each other reveal a quasi-periodic process;

(*iv*) non-stationary processes can present interruption on the lines; this can indicate as well some rare state, or RP fading to the upper left and lower right corners also indicating trend or drifts;

(*v*) single isolate points demonstrate heavy fluctuation in the process, in particular, if only isolate points occur an uncorrelated or anti-correlated random process are represented;

(*vi*) evolutionary processes are illustrated by diagonal lines, then the evolution of states is similar at different times, however, if it has parallel lines related to the main diagonal, the system is deterministic (or even chaotic, if they occur beside single lines), and if the diagonal lines are orthogonal to the main diagonal, or the time is reversed, or the choice of embedding is insufficient;

(*vii*) long bowed line structures express evolution states that are similar at different epochs although they have different velocity (the dynamics of the system could be changing);

(*viii*) vertical and horizontal lines/clusters are evidence that the states have no or slow change for some time, which point to a laminar state.

From Figures 5 and 6, the RPs highlight the recurrences in the signal records showing differences in the dynamical patterns between the HILDCAA interval and the quiet period. To the both systems, the analyses on the large scale patterns in the plots, designated as typology, denote that they are of the disrupted kind, i.e., with abrupt changes in the representation of the dynamics. However, the analysis of the small scale patterns, designated as texture, denotes a more complex dynamics in the HILDCAA event than the one in the quiet interval.

However, the RQA offers a more objective way for the investigation of the considered systems. The density of points, clusters, and structures patterns present in the RP can identify the local behaviour of the analysing data.

**R:** I have a problem with the claim that dynamic properties of HILDCAA-driven activity are "unique". There are distinct differences with the quiet-time study, but uniqueness can be shown only in comparison with other non-HILDCAA driven auroral activity. For instance, analysis of CIR/HSS storms without HILDCAA could be helpful.

**A:** We improved the paper with the extension of data sets concerning the Auroral Electrojet $AE$ index adding other distinct interplanetary conditions. It allows analysis comparisons more complete including, for instance, the suggestion of the Reviewer to analyse CIR/HSS storms without HILDCAAs. The interpretation of "unique event" required clarification. In fact, a modification of the result based on the result of the analysis was necessary. The new information: "The HILDCAAs seem unique events regarding a visible, intense manifestations of interplanetary Alfvenic waves; however, they are similar to the other kinds of conditions regarding a dynamical signature (based on RQA), because it is involved in the same complex mechanism of generating geomagnetic disturbances."

We reformulate the plot of the dynamical comparisons in the paper. The interpretation was reformulated based on a data set extended to our purpose. In the plot attached here as a coloured figure, we present a normalised representation of the RQA parameters for Auroral Electrojet $AE$ indices: in HILDCAA events (○), in CIR/HSS not followed by HILDCAA (x), in a global geomagnetic disturbance scenario (*), and in the geomagnetically quieter intervals (+).

[Figure]

Figure 1: Normalized representation of the RQA parameters for Auroral Electrojet *AE* indices in HILDCAA events (○),in CIR/HSS not followed by HILDCAA (x), in a global geomagnetic disturbance scenario (*), and in the geomagnetically quieter intervals (+).

In short, there are distinct clustering, identified by RQA, of the dynamical behaviours recorded on the ground produced by the interplanetary medium conditions: one regarding a geomagnetically quiet condition regime and another regarding an effective disturbed interplanetary regime. The RQA results identify similar dynamical behaviours for HILDCAA events and the other else disturbed cases.

**R:** I encourage the authors to continue and extend the study with the focus on understanding physical processes behind different types of auroral activity.

**A:** The authors added other data to extend the analysis.

**References**

[revised manuscript text omitted]